# Locating Hazardous Chemical Leakage Source Based on Cooperative Moving and Fixing Sensors

**DOI:** 10.3390/s19051092

**Published:** 2019-03-04

**Authors:** Yaguang Kong, Meng Guan, Song Zheng, Peng Jiang, Ling Wang, Xuefei Yao, Jian Lu, Chenfeng Xie, Fang Wang

**Affiliations:** 1Automation, Hangzhou Dianzi University, Hangzhou 310000, China; ygkong@hdu.edu.cn (Y.K.); moyingcolin@163.com (M.G.); zhs@hdu.edu.cn (S.Z.); HDU_musk@hotmail.com (C.X.); 15958170231@163.com (F.W.); 2Quzhou Juhua Polyamide Fibre LLC, Quzhou 324000, China; jhwlpxl@163.com (L.W.); yaoxuefeiqz@163.com (X.Y.); 13857098555@139.com (J.L.)

**Keywords:** adaptive mutation particle swarm optimization, extended Kalman filtering, cooperative localization, optimized result memory

## Abstract

In dealing with sudden hazardous chemical leakage accidents, the key to solving the evacuation and transfer of personnel and important property is to determine the location of the leakage source and the information of the source strength to gauge the scope of the impact of leakage. The particle swarm optimization algorithm with an adaptive mutation factor is applied to the inverse calculation of leakage source strength to obtain the leakage source information, and the leakage source location problem is transformed into an optimization problem. The mobile sensor is then introduced into the fixed sensor network. The mobile sensor moving strategy based on an extended Kalman filter is proposed. The estimated value of the previous moment and the current time are used to update the estimation of the state variable, and then the mobile strategy is planned. The interference of the random error of the optimization algorithm on the path planning of the mobile sensor is reduced by introducing the optimized result memory and, thus, location efficiency is improved. Simulation results showed that the proposed method, which combines mobile with fixed sensors, greatly expanded the monitoring function of the network, reduced the number of fixed sensors, and enhanced the positioning accuracy.

## 1. Introduction

With the deepening of the industrial revolution, the number of hazardous chemicals has increased dramatically, which has increased the consequent emergencies dramatically. Figure 1 shows the total number of hazardous chemical leakage accidents and the total number of casualties in China from 2010 to 2015 [1].

Cases of heavy losses caused by the leakage of hazardous chemicals are commonplace at home and abroad. For example, at 10:15 on 21 September 2001, an explosion occurred at the AZF chemical plant in Toulouse, an industrial town in southwestern France. The accident killed 31 people and injured more than 2500. On 12 August 2015, the Ruihai Company’s dangerous goods warehouse in the Binhai New Area of Tianjin caught fire in the area, followed by two violent explosions, resulting in 165 deaths, eight missing, and 798 injured.

After a hazardous chemical accident occurs, the information of source strength of the leakage source can determine the dynamic distribution of the leaking material in different space and time positions in the atmosphere. Such information can then be used for early warning, prediction, and emergency treatment of the accident. For many years, various methods have been forwarded and achievements have been made. In 2007, Rao et al. classified source estimation methods into forward model method and backward model method [2]. The backward method is based on reverse propagation and diffusion simulation, i.e., reverse simulation from receiver to source. The forward model method performs multiple transport and diffusion simulations, iteratively calculates from different candidate solutions, and compares the simulation results with the actual measured concentration, in order to find the relevant characteristics of the release source and minimize the difference between the simulated concentration and the actual measured concentration. In 2018, Qiu et al. used artificial neural network to estimate the location of hazardous chemicals leakage source [3]. The method used a large amount of pre-determined scenarios to train the artificial neural network for dispersion prediction, so that the artificial neural network can predict concentration distribution. Combining with recent research results, two ways are used to obtain the source strength information of the leakage source:

The first way is to establish a quantitative estimation model by combining the probabilistic mathematical statistics model with the gas diffusion model, which means performing various characteristics of leakage sources. For example, in 2003, Sohn et al. continuously improved the error estimation by quantifying Monte Carlo sampling with continuous likelihood function, and finally obtained the source strength information of leakage source [4]. In 2018, Kim et al. used a deep neural network to track chemical gas leakage sources [5].

The second way is to establish the relationship between the location and concentration monitoring of the leakage source by using the diffusion law of hazardous chemicals. After which, the concentration calculated by the diffusion model with the concentration value monitored by the sensor can be compared. The optimal solution is obtained by optimization method, while leakage source information can be calculated back. In 2009, Chen Qiang et al. applied the hybrid Genetic-Pattern search algorithm to the inverse calculation of leakage source strength [6]. In 2018, Yan et al. applied the improved Particle Swarm Optimization algorithm to the problem of multi-robot leakage source localization [7].

Generally, source strength location can be divided into two categories, namely, one based on probability and statistics theory and the other based on optimization theory.

The location method based on probability and statistics theory requires much data to be analyzed to solve the problem of fixed sensor network location. Given limited data, the method is not practical. The location method based on optimization theory only needs to provide the concentration measurements at the accident site, which is sui for the case of limited data in emergencies. However, the accuracy of the algorithm should be taken into account while considering calculation efficiency. For example, the genetic algorithm [8] and annealing simulation methods [9] can obtain the global optimal solution of the problem, but the computational efficiency is greatly reduced because of the need to obtain the approximate optimal solution through multiple operations. Nonetheless, several methods can converge quickly, such as traditional particle swarm optimization algorithm, and easily fall into local optimal solution. Subsequently, these methods cannot guarantee global optimal solution, which leads to low positioning accuracy. 

Zhang et al. proposed a self-localization method based on Improved Particle Swarm Optimization in 2011 [10]. The inertia weight and learning factor are set as linear decreasing. Component variation method is used to jump out of local optimum when search stagnation occurs. Although localization performance is improved, the method is still not ideal, given the addition of computational complexity. In 2013, Liu et al. proposed a fast gas leakage source location method based on the combination of real-time monitoring data and Gauss diffusion model from a fixed sensor network monitoring platform [11]. The fixed sensor network mainly relies on a large number of redundant sensors, which require large cost and is not suitable for large-area sites, such as chemical industry parks. In 2015, Jiang et al. proposed the problem of using mobile robots for gas leak source localization [12]. The matrix half-tensor product method was used to plan the robot’s moving path, but the location efficiency was low and the robot was only suitable for small-scale leakage accidents. In 2015, Cheng et al. proposed the combination of wireless sensor networks and mobile robots to solve the problem of gas leakage source location [13]. For the data acquisition method based on the combination of fixed sensor network and mobile sensor, given that the error of the optimization algorithm is derived from random error, the path of the mobile sensor oscillates, which reduces efficiency. 

Based on previous research, this paper proposes to combine the mobile with the fixed sensor networks and use an extended Kalman filter algorithm to update the moving path of the mobile sensor. The particle swarm optimization algorithm of adaptive variation factor is applied to the inverse calculation of the leakage source strength. Using the sum of the squared error of the concentration data calculated by the Gaussian plume diffusion model and the monitoring concentration data as the objective function, the leakage source location is transformed into an optimal matching problem of finding the calculated and the monitoring concentrations. The mobile sensor is introduced in the fixed sensor network and the mobile sensor movement strategy based on the extended Kalman filter is proposed. The estimation of the state variable is updated by using the estimation value of the previous time and the observation value of the current time, and the mobile strategy is planned. Introducing the optimized result memory reduces the interference of the random error of the optimization algorithm on the path planning of the mobile sensor. Subsequently, path oscillation of the mobile sensor is avoided and location efficiency is improved. The simulation results showed that the proposed source-strength location method, which combines mobile and fixed sensors, greatly expanded the monitoring function of the network, improved the effectiveness, accuracy, and flexibility of monitoring, reduced the number of fixed sensors, and improved positioning accuracy.

## 2. Leakage Location of Hazardous Chemicals Based on Adaptive Mutation Particle Swarm Optimization Algorithm

This chapter optimizes the modeling of the leak source, combines the diffusion model with the inversion algorithm, and establishes a leak source location optimization method based on the cooperative positioning network. The particle swarm optimization algorithm is mainly used to estimate the location of the leak source, and the extended Kalman filter algorithm is applied to the planning mobile sensor mobility strategy.

### 2.1. Gas Diffusion Model

To locate the leakage source, the basic law and theory of gas diffusion must be studied to master the characteristics of gas diffusion in space. According to these characteristics, relevant mathematical control methods are used to locate the leakage source of hazardous chemicals [14].

To represent the diffusion form of the leakage material, the diffusion process is described as the fluid movement process. The diffusion model is a Gaussian plume model from a continuous point source at H high above the ground. The equation is as follows:(1)$$C\left(x,y,z\right)=\frac{Q_{0}}{2\pi \sigma_{x}\sigma_{y}u}\exp\left(-\frac{1}{2}\frac{y^{2}}{\sigma_{y}^{2}}\right)\cdot \left\{\exp\left[-\frac{1}{2}\left(\frac{{\left(z-H\right)}^{2}}{\sigma_{z}^{2}}\right)\right]+\exp\left[-\frac{1}{2}\left(\frac{{\left(z+H\right)}^{2}}{\sigma_{z}^{2}}\right)\right]\right\}$$
where:C(*x*,*y*,*z*): concentration of (*x*,*y*,*z*) points;Q: source strength;u: wind speed;*σ_x_*, *σ_y_*, and *σ_z_*: diffusion coefficient in the *x*, *y* and *z* directions, respectively; andH: effective source is high.

In the Gauss diffusion model, atmospheric stability is divided into six grades according to wind speed and sunshine intensity. Table 1 shows the corresponding diffusion coefficient calculation. The value of *σ_x_* is not given because there is reason to assume *σ_x_* = *σ_y_* [15].

The Gauss diffusion model is used to simulate the concentration distribution of hazardous chemicals in windy condition, as shown in Figure 2.

### 2.2. Leakage Source Location Method Based on Self-adaptive Variation Particle Swarm Optimization and Collaborative Location Network

In this paper, the collaborative leakage source localization network of the mobile sensor and fixed sensor network is called the “cooperative localization network”. A fixed sensor network is composed of fixed sensor nodes in the observed area, which can collect, analyze, and process information in the observed area through wireless communication [16]. However, if precise positioning is required, densely fixed sensor networks should be arranged [17]. The introduction of mobile sensors in fixed sensor networks can greatly expand the monitoring function of networks and improve the effectiveness, accuracy, and flexibility of monitoring. The essence of collaborative leakage source localization of the mobile sensor and fixed sensor network is to make up for the shortage of fixed sensor quantity by introducing mobile sensor, cycle iteration, reduce cost, and improve positioning accuracy [18]. 

The coordinate system is established according to the layout of the fixed sensor network. Taking the fixed sensor network with a spacing of 1000 m and a layout of 5 × 5 as an example, the schematic diagram is shown in Figure 3.

Given a certain deviation between the coordinate systems of the fixed sensor network and gas diffusion with the leakage source as the origin, coordinate transformation calculation is needed. The deviation obtained from the two coordinate systems results in the coordinates of the hazardous chemical leakage source in the fixed sensor network coordinate system.

In this paper, the Gauss plume model is used as a forward model to calculate the theoretical value of leakage concentration, that is, the calculated data. The actual concentration value obtained by sensor detection provides monitoring data for the inverse calculation research, and provides data verification for the establishment and correction of diffusion model. Diffusion simulation will also provide simulation data for inverse calculation research. The location of the leak source and its source strength back-calculation will be the best match between the calculated and the monitoring data. By optimizing the solution, the model parameters are continuously corrected to locate the leak source. The determination of the location of the leak source is transformed into a multi-parameter single-object unconstrained optimization problem.

Particle swarm optimization algorithm is applied to the inverse calculation of hazardous chemical leakage source to obtain the data of leakage source rapidly and accurately in the process of chemical leakage accident. As particle swarm optimization algorithm easily falls into local optimal solution, adaptive mutation factor is introduced to avoid this situation [19]. Using a diffusion model and downwind concentration measurement data, the sum of error squared between the calculated concentration data $C_{comp}$ and monitored concentration data $C_{mes}$ is taken as the objective function [20]. That is:(2)$$\min f\left(Q\right)=\sum_{i=1}^{N+t}{\left(C_{mes}^{i}-C_{comp}^{i}\right)}^{2}$$
$C_{comp}$ is derived from Equation (1), that is:(3)$$\min f\left(Q\right)=\sum_{i=1}^{N+t}{\left(C_{mes}^{i}-C_{comp}^{i}\right)}^{2}=\;{\sum_{i=1}^{N+t}\left\{C_{mes}^{i}-\frac{Q}{2\pi \sigma_{x}\sigma_{y}u}\exp\left(-\frac{1}{2}\frac{y^{2}}{\sigma_{y}^{2}}\right)\cdot \left[\exp\left(-\frac{1}{2}\left(\frac{{\left(z-H\right)}^{2}}{\sigma_{z}^{2}}\right)\right)+\exp\left(-\frac{1}{2}\left(\frac{{\left(z+H\right)}^{2}}{\sigma_{z}^{2}}\right)\right)\right]\right\}}^{2}$$
where *N* is the number of fixed sensors in the cooperative localization network and t is the number of steps of the moving sensor.

The process of inverse calculation of source strength using adaptive mutation particle swarm optimization algorithm is as follows:

Initialize the particle swarm, determine the number of particles m, which is the initial velocity and the initial position (that is, the initial parameter value). In this paper, the source strength inversion problem is considered as a three-dimensional space and the parameters are the leakage source position information (x,y) (height is known) and source strength Q, namely, the *i*th particle is represented as $X_{i}=\left(x_{i},y_{i},Q_{i}\right)$ and the velocity is $V_{i}=\left(vx_{i},vy_{i},vQ_{i}\right)$.

Calculate the fitness fit of each particle, store the best values of each particle $P_{best}$ and fit (add one variable), and select the fittest particle from the population as $G_{best}$ of the population:(4)$$V_{i}\left(n+1\right)=wV_{i}\left(n\right)+c_{1}r_{1}\left(P_{best}-X_{i}\left(n\right)\right)+c_{2}r_{2}\left(G_{best}-X_{i}\left(n\right)\right)$$
(5)$$X_{i}\left(n+1\right)=X_{i}\left(n\right)+V_{i}\left(n+1\right)$$

$V_{i}\left(n\right)$ represents the speed of the nth iteration of the *i*th particle, w represents the inertia weight, $c_{1}$ and $c_{2}$ represent the learning parameters, $r_{1}$ and $r_{2}$ represent the random number between 0 and 1, $P_{best}$ represents the optimal value searched by the *i*th particle, $G_{best}$ represents the optimal value searched by the whole cluster, and $X_{i}\left(n\right)$ represents the current position of the nth iteration of the *i*th particle.

According to Equations (4) and (5), the velocity and size of each particle are updated, and the variation factor [21] is introduced to avoid falling into local optimum.

Calculate the fitness of each particle after the update, compare the fit of each particle to the fit corresponding to the best value it has experienced before, and if the result is better, use its current value as the $P_{best}$ of the particle.

Compare the fitness of each particle $P_{best}$ with the best value $G_{best}$ experienced by all particles; if better, the value of $G_{best}$ will be updated.

Determine whether the search results meet the end conditions set by the algorithm (usually to achieve a good enough fitness or to reach the maximum number of iteration steps), if not to meet the preset conditions, then return to step (3). If the preset conditions are met, then stop the iteration and solve the optimal solution.

### 2.3. Mobile Sensor Movement Strategy Planning Based on Extended Kalman Filtering

Kalman filtering is widely used in signal processing and system control, which uses the dynamic information of the target [22], tries to get rid of the influence of noise, and obtain good estimation about the target location. The basic idea is based on minimum mean square error as the best estimate criterion, using the state space model of signal and noise, as well as the estimation of state variables is updated by using the estimated value of the previous moment and the observed value of the current moment, and the estimated value of the current moment is obtained. The algorithm is based on establishing the system of equation and observation equation of the need to deal with the signal to meet the minimum mean square error of the estimate [23].

The state and the observation equations are substituted into the extended Kalman filter equation [24]. The parameters of the filter are determined according to measurement error. The optimal estimation of the position of the mobile sensor can be obtained by starting the filter for recursive calculation and, thus, the trajectory of the mobile sensor can be planned. However, the state and observation equations cannot be all linear. Compared with the Kalman filtering algorithm that is only applicable to linear systems, the extended Kalman filtering algorithm can better plan the mobile sensor’s path [25].

The extended Kalman filtering algorithm is mainly applied to nonlinear systems, which are subsequently linearized, and then the Kalman filtering is carried out [26]. Since the process may be affected by various uncertainties, such as the turbulence of the environment, we assume that the uncertainties of each state component obey normal distribution and are expressed by w and v. The state equation is used to describe the change process of system state over time, which can be expressed as:(6)$$x_{t}=f\left(x_{t-1}\right)+w_{t-1}$$

In the formula, *x_t_* is the state vector of the observed target at step t, $w_{t-1}$ is the process noise, the mean is 0, and the variance is $Q_{t-1}$.

The observation equation is used to describe the observational information of the system state, which is controlled by the nonlinear function h, that is:(7)$$z_{t}=h\left(x_{t}\right)+v_{t}$$

In the formula, $z_{t}$ is the measurement value of the observation target at step t, h is the non-linear function mapping the prediction value to the measurement space, $v_{t}$ is the observation noise, the mean value is 0, and the variance is $R_{t}$.

The extended Kalman filter contains prediction and stages [27]. In the prediction stage, the state equation is used to estimate the state of the system at the next moment, and in the update stage, the observation information at the latest moment is obtained from the observation equation to update the state of the system, which makes the state estimation more accurate.

Prediction stage:(8)$${\hat{x}}_{t|t-1}=F_{t}x_{t-1}$$
(9)$$P_{t|t-1}=F_{t}P_{t-1}F_{t}^{T}+Q_{t}$$

Update stage:(10)$$K_{t}=P_{t|t-1}H_{t}^{T}{\left(H_{t}P_{t|t-1}H_{t}^{T}+R_{t}\right)}^{-1}$$
(11)$${\hat{x}}_{t}={\hat{x}}_{t|t-1}+K_{t}\left(z_{t}-H_{t}x_{t|t-1}\right)$$
(12)$$P_{t}=\left(I-K_{t}H_{t}\right)P_{t|t-1}$$

In the above formulas, $K_{t}$ is called the gain matrix of Kalman filter; $F_{t}$ is the system matrix, which is derived from partial derivative of f; and $H_{t}$ is the Jacobian determinant of a nonlinear system, which is derived from partial derivative of h.

In the location problem of hazardous chemical leakage source, by assuming that the height of the leak source is known, the mobile sensor can be regarded as moving on a two-dimensional plane, and the position coordinates of the mobile sensor in step t are set as $\left(p_{t},q_{t}\right)$. The source strength of the point is derived from Equation (3) of the particle swarm optimization algorithm. The calculated concentration of the position $\left(p_{t},q_{t}\right)$ is $C_{Mcomp}^{t}$ and the measured concentration is $C_{Mmes}^{t}$. The square of the difference between the calculated and the measured concentrations is $C_{Md}^{t}$:(13)$$C_{Md}^{t}={\left(C_{Mmes}^{t}-C_{Mcomp}^{t}\right)}^{2}$$

Definition 1: Let $X_{t}$ be the state vector of the mobile sensor at step t, including the square of the difference between the calculated value of sensor position and concentration and the measured value of concentration, and its expression is $X_{t}=\left[\begin{array}{ccc} p_{t} & q_{t} & C_{Md}^{t} \end{array}\right]$.

Then, the state vector of step t + 1 is:(14)$$X_{t+1}=\left[\begin{array}{c} p_{t+1}\\ q_{t+1}\\ C_{Md}^{t+1} \end{array}\right]=\left[\begin{array}{c} p_{t}+\eta_{p_{t}}\cdot C_{d}^{t}\\ q_{t}+\eta_{q_{t}}\cdot C_{d}^{t}\\ {\left(C_{Mmes}^{t+1}-C_{Mcomp}^{t+1}\right)}^{2} \end{array}\right]+\xi$$
where $\eta_{p_{t}}$ and $\eta_{q_{t}}$ are the moving strategy coefficients of the horizontal and vertical coordinates of the mobile sensor, respectively.

According to the equation of state, the Jacobian matrix is calculated. The diffusion coefficient equation used in $C_{Md}^{t}$ is found in Table 1 according to the atmospheric stability level, that is, the system matrix $F_{t}$ is obtained and its expression is:(15)$$F_{t}=\left[\begin{array}{ccc} \frac{\partial p_{t+1}}{\partial p_{t}} & \frac{\partial p_{t+1}}{\partial q_{t}} & \frac{\partial p_{t+1}}{\partial C_{Md}^{t}}\\ \frac{\partial q_{t+1}}{\partial p_{t}} & \frac{\partial q_{t+1}}{\partial q_{t}} & \frac{\partial q_{t+1}}{\partial C_{Md}^{t}}\\ \frac{\partial C_{Md}^{t+1}}{\partial p_{t}} & \frac{\partial C_{Md}^{t+1}}{\partial q_{t}} & \frac{\partial C_{Md}^{t+1}}{\partial C_{Md}^{t}} \end{array}\right]=\left[\begin{array}{ccc} 1 & 0 & \eta_{p_{t}}\\ 0 & 1 & \eta_{q_{t}}\\ \alpha & \beta & \gamma \end{array}\right]$$
where α, β, and γ are the partial derivatives of $C_{Md}^{t+1}$ with respect to $p_{t}$, $q_{t}$, and $C_{Md}^{t}$, respectively, and can be obtained by Equation (1) and Table 1.

The leakage source information inversely calculated from the co-located network is recorded as $\left(x_{t},y_{t},Q_{t}\right)$, where $\left(x_{t},y_{t}\right)$ is the estimated leak source position coordinate and $Q_{t}$ is the source intensity.

Definition 2: Set the result as the observation vector of mobile sensor in step t, including the coordinate of leakage source position, and its expression is $Z_{t}={\left[\begin{array}{cc} x_{t} & y_{t} \end{array}\right]}^{T}$.

The nonlinear function h of the predicted value mapped to the measurement space can be determined by Equation (2), that is:(16)$$\min f\left(Q\right)=\sum_{i=1}^{N+t}{\left(C_{mes}^{i}-C_{comp}^{i}\right)}^{2}=\;\begin{array}{c} \sum_{i=1}^{N}{\left\{C_{mes}^{i}-\frac{Q}{2\pi \sigma_{x_{i}-x_{t}}\sigma_{y_{i}-y_{t}}}\exp\left(-\frac{1}{2}\frac{{\left(y_{i}-y_{t}\right)}^{2}}{\sigma_{y_{i}-y_{t}}^{2}}\right)\cdot \left[\exp\left(-\frac{1}{2}\left(\frac{{\left(z-H\right)}^{2}}{\sigma_{z}^{2}}\right)\right)+\exp\left(-\frac{1}{2}\left(\frac{{\left(z+H\right)}^{2}}{\sigma_{z}^{2}}\right)\right)\right]\right\}}^{2}\\ +{\left\{C_{mes}^{t}-\frac{Q}{2\pi \sigma_{p_{t}-x_{t}}\sigma_{q_{t}-y_{t}}}\exp\left(-\frac{1}{2}\frac{{\left(q_{t}-y_{t}\right)}^{2}}{\sigma_{q_{t}-y_{t}}^{2}}\right)\cdot \left[\exp\left(-\frac{1}{2}\left(\frac{{\left(z-H\right)}^{2}}{\sigma_{z}^{2}}\right)\right)+\exp\left(-\frac{1}{2}\left(\frac{{\left(z+H\right)}^{2}}{\sigma_{z}^{2}}\right)\right)\right]\right\}}^{2} \end{array}$$
where *N* is the number of groups of all data collected by the collaborative location network before step t of the mobile sensor.

From Equation (16), the relationship between $Z_{t}$ and $X_{t}$ can be obtained, and *h* can be obtained. Hence, the Jacobian determinant $H_{t}$ of nonlinear system can be obtained from the partial derivation of h.

The next position $\left(p_{t+1},q_{t+1}\right)$ of the mobile sensor is planned according to Equations (8) to (12).

### 2.4. Collaborative Leakage Source Location of Mobile Sensor and Fixed Sensor Network

Fixed sensor networks rely mainly on a large number of fixed sensors for positioning and the cost is high. When the mobile sensor is introduced into the fixed sensor network, the number of fixed sensors is reduced and the positioning accuracy is improved.

As the optimization algorithm used in each iteration is a particle swarm optimization algorithm, the result has certain randomness. If the optimization result oscillates, a certain impact will affect the route planning of the mobile sensor. To avoid this situation, the optimization result storage is designed, that is, the result is stored after each iteration. Then, the result of the best matching is selected in the storage, that is, the group with the least fitness as the information of the mobile sensing prediction route. Each iteration selects the optimization result with the least fitness. Hence, as the number of steps increases, the optimization result tends to be the best match, such that the back-calculation result is gradually closer to the real leakage source.

The algorithm for cooperative source location of the mobile sensor and fixed sensor network is as follows:

Step 1: Establish a coordinate system according to the fixed sensor network layout, determine the approximate range of leakage according to the distribution of the fixed sensor reaching the threshold c, and record the moving step of the mobile sensor *t* = 1;

Step 2: Set the particle swarm algorithm to repeat the number of calculations *k* = 1, the maximum value of which is *K*;

Step 3: The collected gas concentration data are located by adaptive mutation particle swarm optimization algorithm, and the result of Equation (3) objective function is recorded as fitness value. Before Step6 starts the mobile sensor, the value of t is 0. After Step 6 starts the mobile sensor, the value of t is determined by Step 8. The optimal solution ${\mathrm{fit}}_{k}$ is obtained by running the particle swarm optimization algorithm, and the leakage source is recorded as $\left(x_{k},y_{k},Q_{k}\right)$, where $x_{k}$ and $y_{k}$ are the x coordinates and y coordinates of the optimal solution of the leakage source obtained by the *k*th particle swarm optimization algorithm, respectively. $Q_{k}$ represents the leakage intensity of the leakage source obtained by the *k*th particle swarm optimization algorithm;

Step 4: If $fit_{k}\geq \varepsilon_{pso}$, then return to Step 3 and run the particle swarm optimization algorithm again. $\varepsilon_{pso}$ is the set inversion calculation accuracy, Otherwise, enter Step 5;

Step 5: Store Step 3 ($x_{k},y_{k},Q_{k},fit_{k}$) in the optimization result store psobest; find the smallest $fit_{k}$ in psobest, denote $fit_{r}$. If $fit_{r}<\varepsilon_{final}$, $\varepsilon_{final}$ represents the final threshold and *ε_final_* < *ε_pso_*, then, the $\left(x_{k},y_{k},Q_{k}\right)$ corresponding to $fit_{r}$ is taken as the leak source position and the leak source intensity, and the method ends. Otherwise, the process proceeds to Step 6;

Step 6: The particle swarm algorithm repeats the number of calculations *k* = *k* + 1. If k<K, return to Step 3, and repeat the particle swarm algorithm, re-execute from Step 3, otherwise, start the mobile sensor and proceed to Step 7;

Step 7: If t<1, then the mobile sensor is placed in the random initial position $\left(p_{0},q_{0}\right)$. Otherwise, the extended Kalman filter algorithm is used to determine the next moving position $\left(p_{t+1},q_{t+1}\right)$ of the mobile sensor. The position coordinate of the mobile sensor in step t is $\left(p_{t},q_{t}\right)$. The calculated value of the concentration is $C_{M\mathrm{co}mp}^{t}$ and the measured value is $C_{M\mathrm{mes}}^{t}$. The square $C_{Md}^{t}$ of the difference between the calculated and the measured values can be calculated by Equation (13). The state vector of step t in mobile sensor can be expressed as $X_{t}={\left[\begin{array}{ccc} p_{t} & q_{t} & C_{Md}^{t} \end{array}\right]}^{T}$. Step 5 obtains the coordinate $\left(x_{r},y_{r}\right)$ of the leakage source, and the observation vector of the mobile sensor’s step t can be expressed as $z_{t}={\left[\begin{array}{cc} x_{r} & y_{r} \end{array}\right]}^{T}$. The extended Kalman filter algorithm, i.e., Equations (8) to (16) are used to plan the mobile sensor’s moving strategy to estimate the next position $\left(p_{t+1},q_{t+1}\right)$ of the mobile sensor.

Step 8: Calculate the theoretical concentration value $C_{Mcomp}^{t+1}$ at the position of $\left(p_{t+1},q_{t+1}\right)$; move the mobile sensor to $\left(p_{t+1},q_{t+1}\right)$ and measure the coordinate concentration to obtain the actual concentration value $C_{Mmes}^{t+1}$; and move the sensor step t = t + 1;

Step 9: If $C_{Mmes}^{t+1}\geq c$, that is, the point is in the leakage range. Calculate $C_{d}^{t}=\sum_{i=1}^{N+t}{\left(C_{mes}^{i}-C_{comp}^{i}\right)}^{2}$, where N is the cooperative localization network collected before step t of the mobile sensor. If $C_{d}^{t}<\varepsilon_{final}$, the position $(p_{t+1},q_{t+1})$ is used as the leak source position, and the method is terminated, otherwise, record $\left(p_{t+1},q_{t+1},C_{Mmes}^{t+1}\right)$ as the new data collected by the mobile sensor, and jump to Step 2 to execute. If $C_{Mmes}^{t+1}<c$, return to Step 2 to execute again.

The flowchart of the algorithm is shown in Figure 4.

## 3. Simulation and Analysis

In order to verify the feasibility of the algorithm in the problem of locating leakage sources, assuming that the source strength is Q = 10,000 g/s, the atmospheric stability is C, and the average wind speed is 2.5 m/s, the simulated measured concentration values at different (*x*, *y*, *z*) locations are generated according to Equation (1). The feasibility of the algorithm is verified by using this data.

### 3.1. Influence of Factors in Fixed Sensor Networks on Location Accuracy of Leakage Source

The influence of factors, such as sensor spacing, sensor layout, and layout on the positioning accuracy in fixed sensor networks is important. The error formula is:(17)$$e=\sqrt{{\left(x-x_{0}\right)}^{2}+{\left(y-y_{0}\right)}^{2}}$$
*e*: error distance;*x*: abscissa of the leak source obtained from the experimental results;*y*: ordinate of the leak source obtained from the experimental result;*x*_0_: actual leak source abscissa; and*y*_0_: actual leak source ordinate.

The leakage source coordinate is assumed to be (650,400) regardless of the height, and 100 sets of experiments are performed in each case.

The first set of experiments discussed the same layout, that is, the layout is 9 × 9 fixed sensor network, and the impact of sensor spacing on positioning accuracy.

The sensor spacings are 10, 50, 100, 500, 1000 and 2000 m, and the error is shown in Figure 5.

When the layout is the same, the average positioning error of different sensor intervals is shown in Table 2.

The second set of experiments explores the effect of layout on positioning accuracy in the case of the same sensor spacing.

The sensor layout is taken as 3 × 3, 4 × 4, 5 × 5 and 9 × 9. The positioning accuracy of the sensor spacing is 1000 and 2000 m. The error is shown in Figure 6 and Figure 7.

The average positioning error of different layouts is shown in Table 3 when the sensor spacing is the same.

The experimental results show that the larger the distribution range of the fixed sensor networks is under the same sensor spacing, the better the positioning effect is when the number of fixed sensors is greater. The smaller the sensor spacing is, the better the positioning accuracy. With the increase of the distance between sensors in fixed sensor networks, the location error will increase as well. To obtain relatively accurate positioning results, increasing the distribution density of sensors, such as 10 m, is necessary. However, this scheme will increase the cost for chemical industry parks.

### 3.2. Effect of Coordinated Leakage Source Location on Positioning Accuracy Between the Mobile Sensor and Fixed Sensor Network

As the error of the optimization algorithm is derived from random error, the path of the mobile sensor oscillates. Hence, the introduction of optimization result memory will reduce the interference of the random error of the optimization algorithm on the planned path of the mobile sensor and avoids the oscillation of the path of the mobile sensor, thereby improving positioning efficiency. The following is an example of introducing a motion sensor cooperative positioning experiment in a fixed sensor network with a layout of 8000 m × 8000 m, a sensor spacing of 1000 m, and a layout of 9 × 9. The path of the leakage source is (650,400). The path of the mobile sensor is shown in Figure 8, where (a) is the path of the moving sensor before introducing the optimization result memory and (b) is the path of the moving sensor after introducing the optimization result memory, where red is the true coordinate of the leak source and blue is the moving path of the moving sensor.

To represent more intuitively the moving steps of the mobile sensor, the variable step t is introduced into the path image of the mobile sensor, as shown in Figure 9.

Under the condition of introducing mobile sensors, the influence of factors, such as sensor spacing, sensor layout range, and layout on the positioning accuracy in fixed sensor networks are discussed.

The leakage source coordinate is assumed to be (650,400) regardless of the height and 100 sets of experiments are performed in each case.

The first group discusses the positioning error of cooperative location networks with different layouts when the sensor spacing is 1000 m.

The sensor layout is 3 × 3, 4 × 4, 5 × 5 and 9 × 9, and the error is shown in Figure 10.

The second group discusses the positioning error of cooperative location networks with different layouts when the sensor spacing is 2000 m.

The sensor layout is 3 × 3, 4 × 4, 5 × 5 and 9 × 9 and the error is shown in Figure 11.

The comparison of the positioning error results of different sensor networks is shown in Table 4.

The comparison of the positioning accuracy between the fixed sensor network and the collaborative location network is shown in Figure 12.

Experimental results show that the positioning accuracy is greatly improved after the introduction of the mobile sensor. The coordinated source location method of the mobile and the fixed sensor networks improves accuracy of the leak source localization result. The use of a fixed sensor network combined with a mobile sensor can effectively obtain accident information, make up for the shortcomings of relying solely on fixed sensor network monitoring to cause malfunction, and improve the flexibility of accident monitoring. The introduction of a mobile sensor reduces the distribution density of fixed sensors in a fixed sensor network. For example, in a 16 km^2^ sensor network, the positioning accuracy of the collaborative location network with 9 fixed sensors spacing 2000 m is 25.79% higher than that of the fixed sensor network with 25 fixed sensors spacing 1000 m, and the number of fixed sensors is reduced by 16. In a 64 km^2^ sensor network, 25 fixed sensors are used. The positioning accuracy of collaborative location network with sensor spacing of 2000 m is 55.89% higher than that of 81 fixed sensor networks with fixed sensor spacing of 1000 m, and the number of fixed sensors is reduced by 56. The collaborative location network combined with mobile sensor and fixed sensor network can reduce the number of fixed sensor distribution, thus greatly reducing the cost.

### 3.3. Running Time Statistics of the Algorithm

Since the mobile sensor in this paper is regarded as the theoretical state, that is, there is no speed limit, the moving time of the mobile sensor is not added in the running time of the algorithm. The average running time of the collaborative location network positioning algorithm for different layouts is shown in Figure 13.

As shown in the figure, for sensor networks with different layouts, the average running time of the algorithm is less than 1 min, which ensures fast location of the leakage source.

### 3.4. Comparison with other Algorithms

We compared the Random forest classifier for real-time chemical leak source tracking using fence-monitoring sensors method [28]. Random forest (RF) classification is an ensemble learning method for classification and other tasks. It is operated by constructing a multitude of decision trees using training data and outputting the results.

When the chemical leak occurs, 11 sensors which are optimally placed on the fence of the plant detect the leak. The data was split in a 7:3 ratio for the training and test datasets. In Model 1, 50 trees were used in the forest with Gini criterion (data split method). It was designed with 13 features and 40 labeled data (40-class). After training the model, the error rate was 58.04%. Model 2 was designed by adding one more class, which is the no leak release class, the error rate 46.75%. Model 3, which was designed with 90 features and 40-class, the error rate 14.01%. Model 4 gave the maximum accuracy when the optimal tuned parameters are being used: 400 trees, 20% maximum-feature, and one minimum-sample-leaf, the error rate was 13.15%.

We use the collaborative location network with a distance of 1000 m as a comparison. When the layout is 3 × 3, the error rate is 11.95%. When the layout is 4 × 4, the error rate is 2.16%. When the layout is 5 × 5, the error rate is 1.26%. When the layout is 9 × 9, the error rate is 0.92%.

It can be seen that locating hazardous chemical leakage source based on cooperative moving and fixing sensors has good positioning accuracy while ensuring efficiency.

## 4. Conclusions

Aiming at the problem of hazardous chemical leakage source location, this paper studies the existing diffusion mode and the model method of source strength inversion and location. A mobile sensor and a fixed sensor network based on the extended Kalman filter are proposed to locate a hazardous chemical leakage source, and the feasibility of the algorithm is verified. The particle swarm optimization algorithm with adaptive mutation is applied to the source strength inversion problem. Subsequently, the source strength inversion problem of hazardous chemicals leakage is transformed into the traditional optimization model. A mobile sensor mobile strategy based on extended Kalman filtering is then proposed. Due to the randomness of the results obtained by the particle swarm optimization algorithm, the calculated path of the mobile sensor will oscillate, resulting in the long path of the mobile sensor and inaccurate location. To improve this situation, an optimal value memory is designed. After adding the optimal value memory, the route oscillation of the mobile sensor is obviously improved. Compared with values before the improvement, the mobile path of the mobile sensor is shortened, which can locate the location of the leakage source quickly and accurately. Thus, the improved working efficiency of the mobile sensor saves time and cost and is more suitable for the needs of emergency response process. The cooperative positioning scheme of the mobile sensor and fixed sensor network can improve effectiveness, accuracy, and flexibility of monitoring. The scheme can quickly and accurately obtain source strength information and provide support for emergency rescue. The collaborative location network, which combines mobile with fixed sensor networks, can reduce the number of fixed sensors and, thus, greatly reduces cost.

Given that the leak model used in the algorithm is the plume model of a continuous point source, the meteorological environment is relatively stable. However, real-time meteorological changes are not considered, which makes the algorithm unsuitable for an environment with large meteorological changes. In the follow-up study, a time factor can be added to guide the mobile sensor through real-time updating data, which could ensure that effective concentration information can be monitored in the case of large changes in meteorological environment, and provide sufficient data support for the leak source location.

By comparing the positioning accuracy of fixed sensor networks with that of a collaborative location network, when sensors are enough in fixed sensor networks, such as 9 × 9 distribution, the positioning accuracy of cooperative positioning networks with mobile sensors is not much improved compared with that of the 5 × 5 distribution, indicating that they are not fixed sensors. The more sensors in the network, the better. In the follow-up research, we can study the number of fixed sensors in the collaborative location network and find the minimum number of fixed sensors while ensuring positioning accuracy.

## Figures and Tables

**Figure 1 sensors-19-01092-f001:**
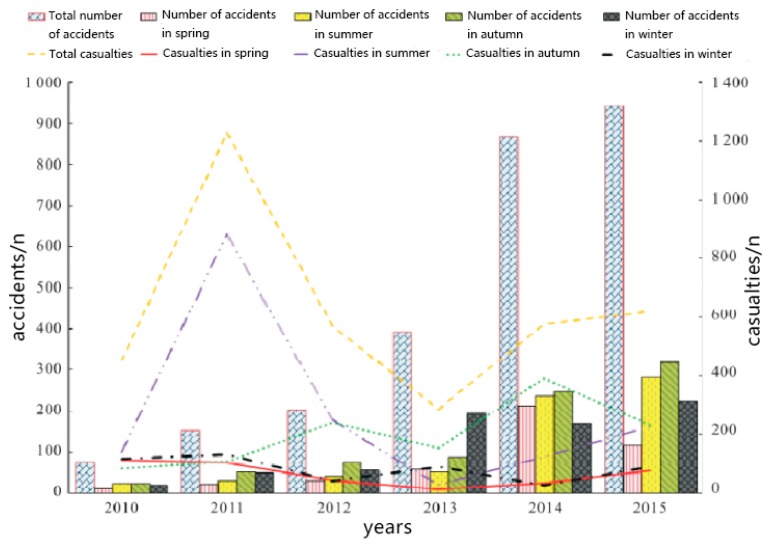
Total number of deaths and total casualties of hazardous chemicals spills in China from 2010 to 2015.

**Figure 2 sensors-19-01092-f002:**
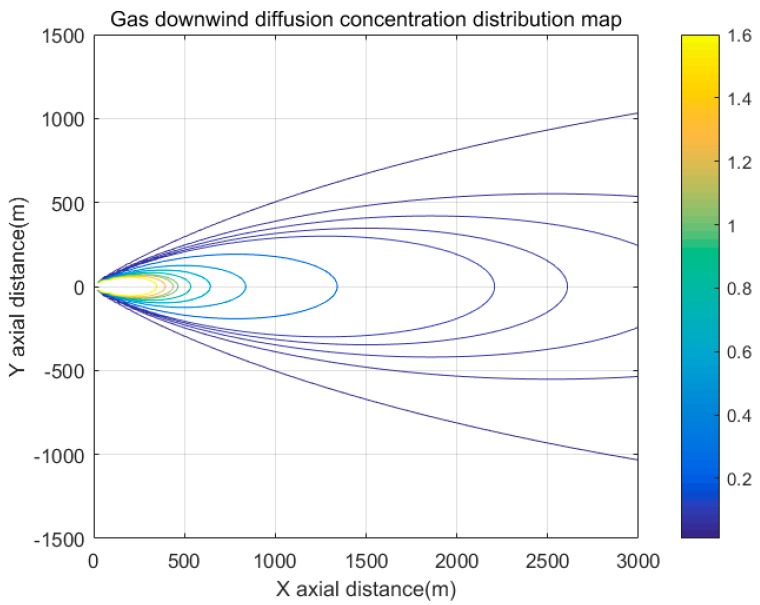
Concentration distribution of hazardous chemicals after leakage in windy conditions.

**Figure 3 sensors-19-01092-f003:**
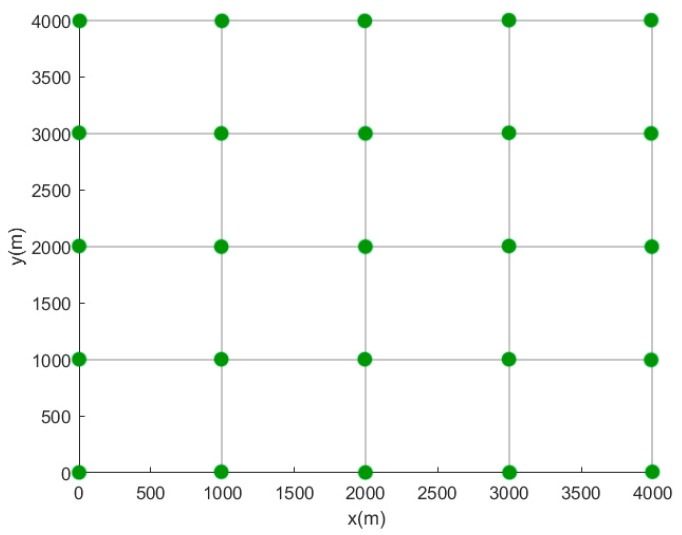
Schematic diagram of establishing coordinate system for the fixed sensor network.

**Figure 4 sensors-19-01092-f004:**
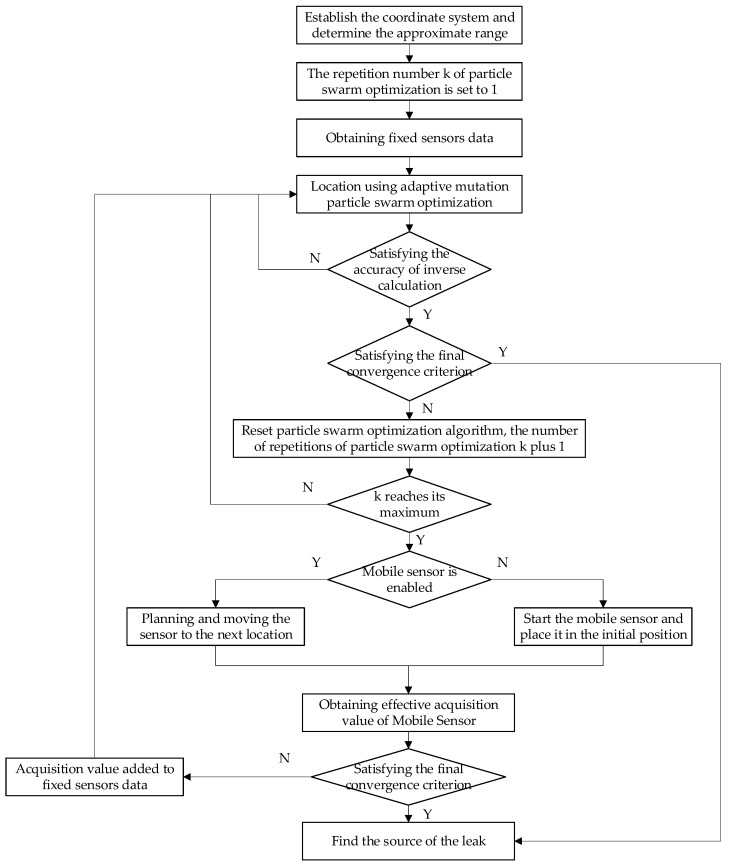
Flowchart of collaborative leak source localization algorithm for mobile sensor and fixed sensor network.

**Figure 5 sensors-19-01092-f005:**
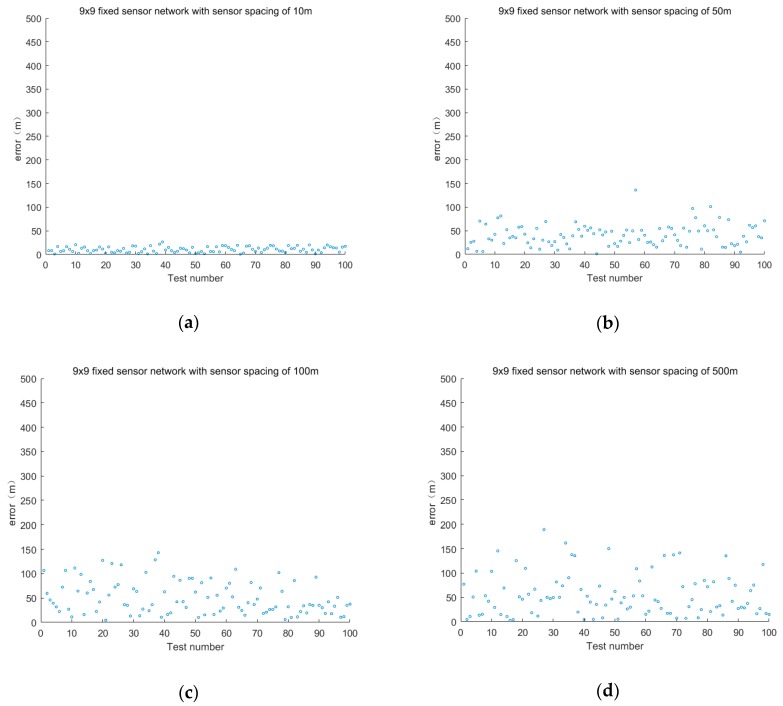
Positioning the error of different sensor spacings with the same layout.

**Figure 6 sensors-19-01092-f006:**
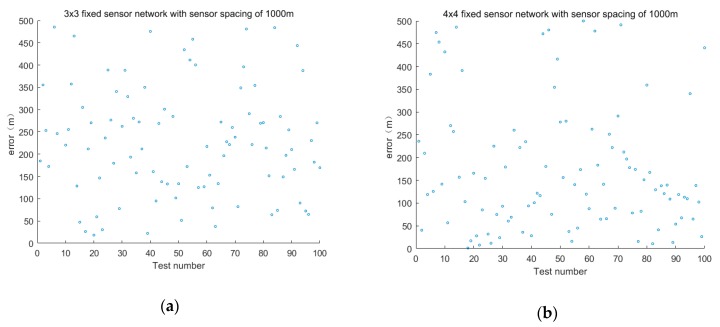
Positioning error of fixed sensor networks with different layouts at 1000 m intervals.

**Figure 7 sensors-19-01092-f007:**
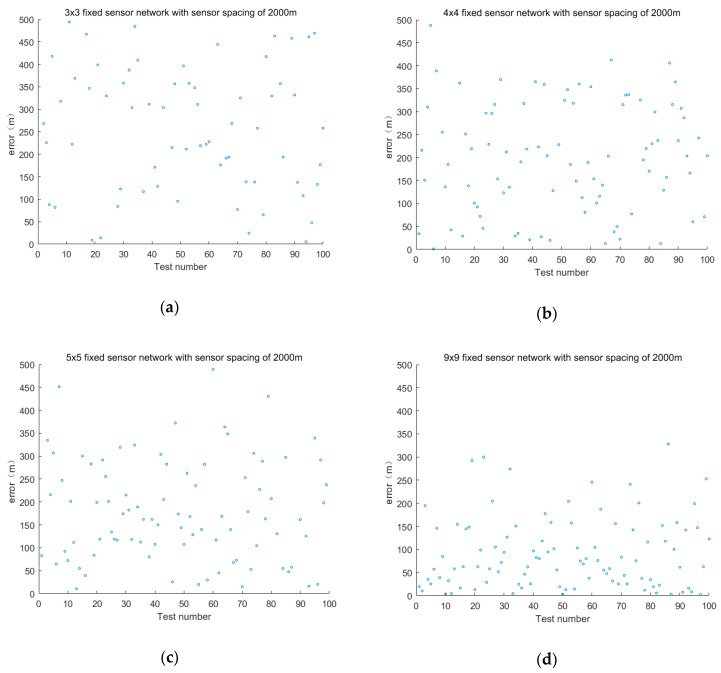
Positioning error of fixed sensor networks with different layouts at 2000 m intervals.

**Figure 8 sensors-19-01092-f008:**
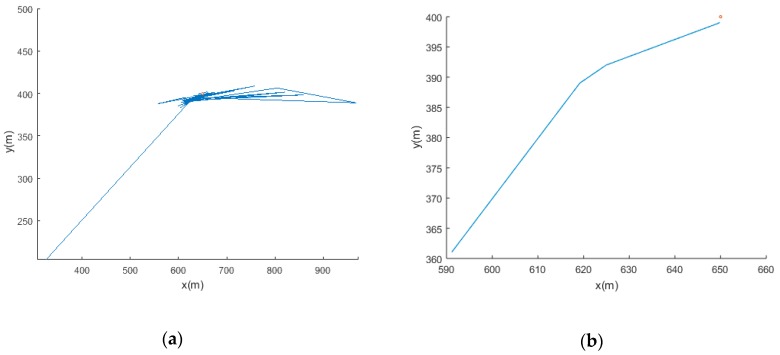
Mobile sensor path (**a**) before introducing the optimization result memory; and (**b**) after introducing the optimization result memory.

**Figure 9 sensors-19-01092-f009:**
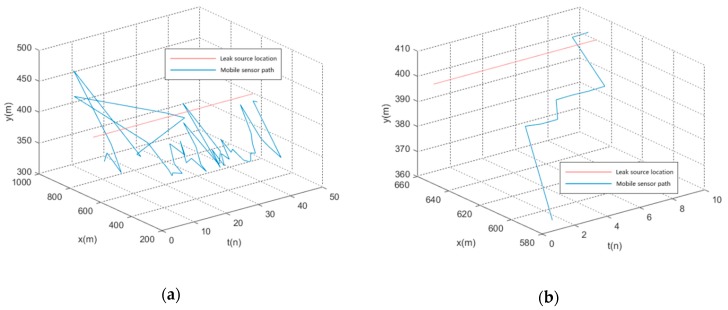
Mobile sensor moving steps (**a**) before introducing the optimization result memory; and (**b**) after introducing the optimization result memory.

**Figure 10 sensors-19-01092-f010:**
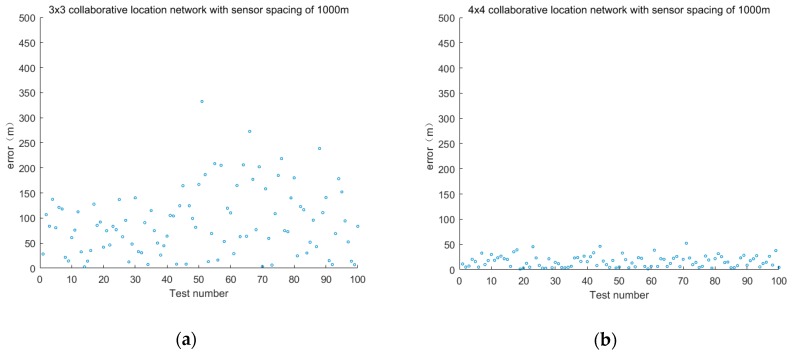
Positioning error of cooperative location network with different layouts of sensors at 1000 m intervals.

**Figure 11 sensors-19-01092-f011:**
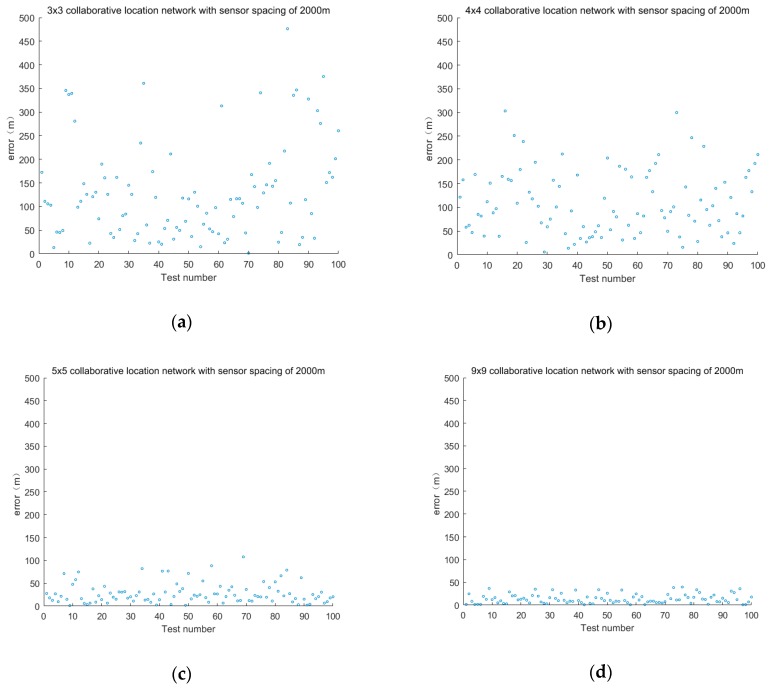
Positioning error of cooperative location network with different layouts of sensors at 2000 m intervals.

**Figure 12 sensors-19-01092-f012:**
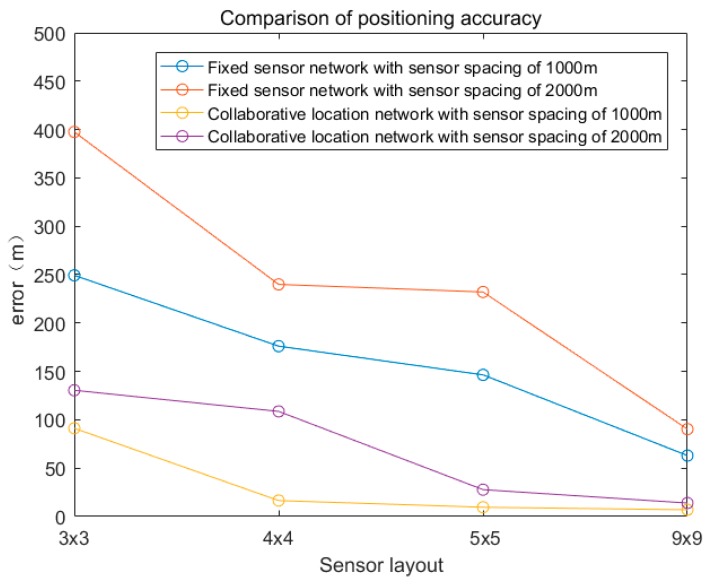
Comparison of positioning accuracy between the fixed sensor network and collaborative location network.

**Figure 13 sensors-19-01092-f013:**
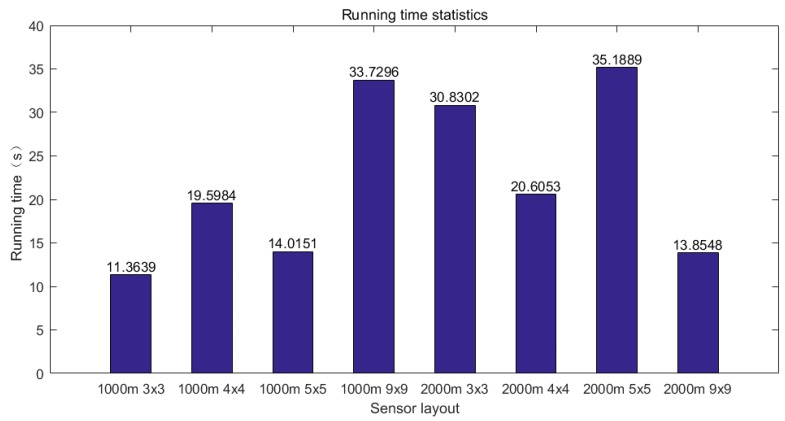
Statistics of the average running time of the algorithm.

**Table 1 sensors-19-01092-t001:** Diffusion coefficient equation of the Gauss diffusion model.

Stability Grade of Gauss Diffusion Model	*σ_y_*/*m*	*σ_z_*/*m*
Rural conditions		
A	0.22 × (1 + 0.0001×)^−1/2^	0.20×
B	0.16 × (1 + 0.0001×)^−1/2^	0.12×
C	0.11 × (1 + 0.0001×)^−1/2^	0.08 × (1 + 0.0002×)^−1/2^
D	0.08 × (1 + 0.0001×)^−1/2^	0.06 × (1 + 0.0015×)^−1/2^
E	0.06 × (1 + 0.0001×)^−1/2^	0.03 × (1 + 0.0003×)^−1/2^
F	0.04 × (1 + 0.0001×)^−1/2^	0.016 × (1 + 0.0003×)^−1/2^
Urban conditions		
A–B	0.32 × (1 + 0.0004×)^−1/2^	0.24 × (1 + 0.0001×)^−1/2^
C	0.22 × (1 + 0.0004×)^−1/2^	0.20 ×
D	0.16 × (1 + 0.0004×)^−1/2^	0.14 × (1 + 0.0003×)^−1/2^
E–F	0.11 × (1 + 0.0004×)^−1/2^	0.08 × (1 + 0.0015×)^−1/2^

**Table 2 sensors-19-01092-t002:** Average positioning error of different sensor intervals with the same layout.

Sensor Layout	Sensor Spacing (m)	Sensor Layout Range (m × m)	Fixed Sensor Network Positioning Error (m)
9 × 9	10	80 × 80	10.49
	50	400 × 400	40.63
	100	800 × 800	50.39
	500	4000 × 4000	55.16
	1000	8000 × 8000	63.01
	2000	16,000 × 16,000	90.30

**Table 3 sensors-19-01092-t003:** Average positioning error of different layouts with the same sensor interval.

Sensor Spacing (m)	Sensor Layout	Sensor Layout Range (m × m)	Fixed Sensor Network Positioning Error (m)
1000	3 × 3	2000 × 2000	249.32
	4 × 4	3000 × 3000	176.09
	5 × 5	4000 × 4000	146.45
	9 × 9	8000 × 8000	63.01
2000	3 × 3	4000 × 4000	397.55
	4 × 4	6000 × 6000	239.94
	5 × 5	8000 × 8000	232.09
	9 × 9	16,000 × 16,000	90.30

**Table 4 sensors-19-01092-t004:** Test positioning error results of different sensor networks.

Sensor Spacing (m)	Sensor Layout	Sensor Layout Range (m × m)	Fixed Sensor Network Positioning Error (m)	Collaborative Location Network Positioning Error (m)	Positioning Accuracy Improvement Rate
+1000	3 × 3	2000 × 2000	249.32	91.22	63.41%
	4 × 4	3000 × 3000	176.09	16.50	90.62%
	5 × 5	4000 × 4000	146.45	9.60	93.44%
	9 × 9	8000 × 8000	63.01	7.01	88.87%
2000	3 × 3	4000 × 4000	397.55	130.48	67.17%
	4 × 4	6000 × 6000	239.94	108.67	54.70%
	5 × 5	8000 × 8000	232.09	27.79	88.02%
	9 × 9	16,000 × 16,000	90.30	13.97	84.52%

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
