# Peer review of "Locating Hazardous Chemical Leakage Source Based on Cooperative Moving and Fixing Sensors"

_sensors, 2019, doi:10.3390/s19051092_

Round 1

Reviewer 1 Report

The manuscript uses a combination of a simplified diffusion model, a particle swarm optimization algorithm, and Extended Kalman filtering to plan for the motion of a mobile sensor searching for a discharge of, in their case, a hazardous chemical substance. 

The strategy outlined in the document has relevance to many areas where fixed monitoring networks are being used to detect, or warn about, an adverse event, and a moving observer is subsequently launched of verification and localization. 

The text is in some parts hard to read. Especially the algorithms contain a lot of text, occasionally repeating what has already been said. So I would suggest to request a thorough rewriting of the manuscript for more clarity. 

My main concern about the suggested strategy is the need for using the swarm optimizing to estimate the location of the discharge. Can’t the Kalman filter, based on the simplified diffusion model and the measurements, give that estimate? 

My second concern is the very simplified diffusion model being used as the forward modell for the Kalman filter. It is well known that the environment will be turbulent and hence this model provide a too smooth signal from a discharge. The Kalman filter is designed to take such model uncertainties into account. So I suggest the authors at least discuss that, and maybe bring that into the equations. 

Since the manuscript addresses a general problem, the reference list could be more extensive. A quick web search came up with these publications, that might be of interest to the authors: 

Karaman, S., and E. Frazzoli (2011), Sampling-based algorithms for optimal motion planning, Int. J. Robotics Res.30(7), 846–894, doi: 10.1177/0278364911406761. 

Breivik, Ø., A. A. Allen, C. Maisondieu, and M. Olagnon (2012), Advances in search and rescue at sea, Ocean Dyn.63(1), 83–88, doi:10.1007/ s10236-012-0581-1. 

Hollinger, G. A., and G. S. Sukhatme (2014), Sampling-based robotic information gathering algorithms, Int. J. Robotics Res.33(9), 1271– 1287, doi:10.1177/0278364914533443. 

Stone, L. D., C. M. Keller, T. M. Kratzke, and J. P. Strumpfer (2014), Search for the Wreckage of Air France Flight AF 447, Stat. Sci.29(1), 69–80, doi:10.1214/13-STS420. 

Alendal, G. (2017). Cost efficient environmental survey paths for detecting continuous tracer discharges. Journal of Geophysical Research-Oceans122(7), 5458–5467. http://doi.org/10.1002/2016jc012655

Hence, my recommendation is to request a major revision to make the strategy clearer and the manuscript easier to read. There is also a number of typos in the manuscript and I will encourage the authors to take a closer look at their equations. For instance, there is no explicit dependency on x in equation 1, what is C_d^t in equation 14, and should the y_j in the second line of Eq (16) be p_t?  

All in all an interesting manuscript, but it should be rewritten for clarity. 

Author Response

The algorithm description has been modified.

The algorithm mainly uses the adaptive particle swarm optimization algorithm to locate the source of hazardous chemicals leakage, and the extended Kalman filter algorithm is used to plan the moving path of the mobile sensor. The Kalman filter algorithm is only applicable to linear systems. The observation vector expression in this algorithm is determined by the inverse calculated position coordinates of the particle swarm algorithm, and belongs to the nonlinear system. Therefore, the extended Kalman filter algorithm suitable for nonlinear systems is used to plan the moving path of the mobile sensor.

The state equations and observation equations in the Kalman filter algorithm contain noise terms w and v, respectively, which mean the influence of various uncertain factors that may be affected, including turbulence in the environment. An explanation of the noise term is added in Section 2.3.

Formula (1) does not explicitly depend on x, but Table 1 shows that the values of σx , σy and σz are all related to x.

C_d^t in the formula (14) can refer to the formula (13), which means the square of the difference between the concentration calculation value and the concentration measurement value of the position of the mobile sensor at the step t.

Formula (16) is used to find the non-linear function h in which the predicted value is mapped to the measurement space. The x_t and y_t in the formula are taken from the observation vector expressions Z_t, x_i and y_i (i=1,2,... N+t represents all the valid data collected by the collaborative location network (i.e., fixed sensors and mobile sensors).

Reviewer 2 Report

In order to locate the hazardous chemicals leakage source, this paper introduces a mobile sensor into the fixed sensor network with extended Kalman filter method. Then, authors use the particle swarm optimization algorithm with adaptive mutation to do the inversion calculation of source strength. This paper shows a lot of work the authors have done. However, there are some problems within the algorithm part and the experiment part.

1.The last paragraph of Section 1 “Introduction” (Line 65-106) is too long to read. It is suggested to highlight the main contributions of this paper by items at the end of section 1.

2. Section 2 introduces the entire algorithm, but I cannot sort out the logical ideas of this section by the titles of subsections. Maybe authors should add some descriptions of logical structure of section 2.

3. What does the variables “Cmes” and “Ccomp” mean in equation 2. Moreover, it is stated in line 169 that “T is the number of steps of the moving sensor. ”, but I just find the variable “t” in equation 2. Please check the case of this variable.

4. Equation 3 defines the objective function of particle swarm optimization algorithm. From this equation, concentration data from measurement and diffusion model are both needed. However, section 3 just performs the simulations and how can you simulate the real data as to confirm the validity of the algorithm.

5. Figure 4 and Figure 5 are unnecessary in this paper. On the contrary, authors can combine these two figures into a flowchart of general design.

6. It is not convincing that there are no comparison simulations between designed algorithm and other advanced method.

7. The evaluation index of simulations is only the “error”. However, I think it is more important to analyze the running time of entire algorithm for locating the hazardous chemicals leakage source.

8. Section 3 discussed the influence factors (sensors spacing and layout) in different sensor networks.

9. The amount of references are not enough to express authors’ ability of literature review.

Author Response

1.The 65-106 paragraph is divided into three paragraphs, 65-75 analyzes the inadequacies of the two methods, 76-91 is the research progress of the predecessors, 92-108 is the basic idea of this paper.

2.In the second chapter, the introduction (111-115) is introduced: This chapter optimizes the modeling of the leak source, combines the diffusion model with the inversion algorithm, and establishes a leak source location optimization method based on the cooperative positioning network. The particle swarm optimization algorithm is mainly used to estimate the location of the leak source, and the extended Kalman filter algorithm is applied to the planning mobile sensor mobility strategy.

3.The meanings of "Cmes" and "Ccomp" have been added, and "T" has been changed to "t".

4.An introduction is given in Chapter 3 to describe how to simulate data.

5.Figure 4 and Figure 5 have been deleted and a new flowchart has been added.

6.7.8.9.Added comparison with advanced algorithms, added algorithm runtime results, and increased the number of references.

Round 2

Reviewer 2 Report

I have no other queations.

Author Response

Bibliography has extended.